# Fisetin, a Potent Anticancer Flavonol Exhibiting Cytotoxic Activity against Neoplastic Malignant Cells and Cancerous Conditions: A Scoping, Comprehensive Review

**DOI:** 10.3390/nu14132604

**Published:** 2022-06-23

**Authors:** Robert Kubina, Kamil Krzykawski, Agata Kabała-Dzik, Robert D. Wojtyczka, Ewa Chodurek, Arkadiusz Dziedzic

**Affiliations:** 1Department of Pathology, Faculty of Pharmaceutical Sciences in Sosnowiec, Medical University of Silesia, 30 Ostrogórska Str., 41-200 Sosnowiec, Poland; adzik@sum.edu.pl; 2Silesia LabMed: Centre for Research and Implementation, Medical University of Silesia in Katowice, 18 Medyków Str., 40-752 Katowice, Poland; kamil.krzykawski@sum.edu.pl; 3Department of Microbiology and Virology, Faculty of Pharmaceutical Sciences in Sosnowiec, Medical University of Silesia in Katowice, 4 Jagiellońska Str., 41-200 Sosnowiec, Poland; rwojtyczka@sum.edu.pl; 4Department of Biopharmacy, Faculty of Pharmaceutical Sciences in Sosnowiec, Medical University of Silesia in Katowice, 8 Jedności Str., 41-208 Sosnowiec, Poland; echodurek@sum.edu.pl; 5Department of Conservative Dentistry with Endodontics, Medical University of Silesia, 17 Akademicki Sq., 41-902 Bytom, Poland; adziedzic@sum.edu.pl

**Keywords:** anticancer, flavonoid, flavonol, fisetin, cancer

## Abstract

Diet plays a crucial role in homeostasis maintenance. Plants and spices containing flavonoids have been widely used in traditional medicine for thousands of years. Flavonols present in our diet may prevent cancer initiation, promotion and progression by modulating important enzymes and receptors in signal transduction pathways related to proliferation, differentiation, apoptosis, inflammation, angiogenesis, metastasis and reversal of multidrug resistance. The anticancer activity of fisetin has been widely documented in numerous in vitro and in vivo studies. This review summarizes the worldwide, evidence-based research on the activity of fisetin toward various types of cancerous conditions, while describing the chemopreventive and therapeutic effects, molecular targets and mechanisms that contribute to the observed anticancer activity of fisetin. In addition, this review synthesized the results from preclinical studies on the use of fisetin as an anticancer agent. Based on the available literature, it might be suggested that fisetin has a bioactive potential to become a complementary drug in the prevention and treatment of cancerous conditions. However, more in-depth research is required to validate current data, so that this compound or its derivatives can enter the clinical trial phase.

## 1. Introduction

Cancer incidence is continuously increasing worldwide. New Globocan 2020 cancer data covering 185 countries revealed over 19 million new cancer cases and over 9.8 million cancer-related deaths worldwide for both genders of all ages. The most common, in general, are breast cancer (11.7%), lung cancer (11.4%) and colorectal cancer (10%). Right behind these, in the top ten are prostate, stomach, liver, cervical, esophagus, thyroid and bladder cancer [1]. A significant discrepancy exists when distinguishing gender. The first three places in men are followed by lung, prostate and intestine cancer, and in women these are breast, intestine and lung cancer [2]. It is precisely due to the increased morbidity and mortality from cancer that early prevention in the form of proper nutrition has become extremely important. As we know, insufficient and improper diet is the second element of our everyday life aspects, after smoking, that influences the risk of cancer [3]. Scientists from all over the world are intensively searching for measures to minimize the risk of cancerous condition development. As the oncology knowledge in this field is becoming more complete, and equally multifactorial and complex, it has been established that the balanced, healthy diet, maintaining a normalized body weight and physical activity protect against various forms of malignant tumors; among others, these reduce the risk of esophageal cancer by 69%; the risk of mouth, pharynx and larynx cancer by 63%, the risk of uterine cancer by 59%; the risk of colorectal cancer by 50%; the risk of stomach cancer by 47%; the risk of breast cancer by 38%; and the risk of colon and rectal cancer by 75% [4].

Numerous in vitro and in vivo studies have confirmed that natural phytochemicals such as plant-derived alkaloids and flavonoids exhibit substantial anticancer activity against various cancerous cells. The broad range of their potential applications in biomedical research and clinical practice justifies further well-designed studies assessing the multidirectional pathways of phytochemical anticancer activities, particularly in combination with standard chemotherapeutics. Such research requires a detailed evaluation of natural compound benefits vs. their potential negative impact on biological processes in humans. It is expected that, in the future, the standardized therapeutic protocols in oncology will allow the use of multiple naturally originating compounds to enhance clinical outcome and reduce side effects of anticancer therapies. While the magnitude of synergistic activity of plant-derived flavonoids and chemotherapeutics remains unclear, there is an enormous potential for the future research providing rational, evidence-based data for the oncology field. The hidden potential and magnitude of natural compounds against cancer cells is yet to be discovered, based on robust and well-designed studies with replicable results. Interestingly, it is worth noticing that many current first-line anticancer drugs such as paclitaxel and colchicine originate from natural products [5,6].

### Flavonols

Flavonols (FVLs, IUPAC name 3-hydroxy-2-phenylchromen-4-one), plant secondary metabolites, is a subgroup of flavonoid organic compounds. FVLs, including fisetin, are derivatives of flavonol, with each compound in this group containing a hydroxyl group (-OH) on the third carbon atom (C3) and a carbonyl group (C=O) on the fourth carbon atom of the central heterocyclic ring (Figure 1) [7]. The diversity of FVLs is due to the location of additional -OH groups in the molecule and sometimes additional methyl groups.

#### Fisetin—General Characteristics

Fisetin (3,3′,4′,7-tetrahydroxy flavone, FIS) is a naturally occurring flavonoid found in various fruits (mangoes, apples, strawberries, kiwis and grapes), vegetables (cucumbers, tomatoes and onions), nuts and wine [8], which show strong anti-inflammatory [9], antioxidant [10], anticancer [11], anti-invasive, anti-angiogenic [12], antidiabetic [13], neuroprotective and cardioprotective activity [14]. FIS is currently sold and available mainly as a supplement containing *Cotinus coggygria* extract. Long-term oral FIS supplementation is expected to cause few side effects at a daily dose of 200 mg. FIS has recently gained attention in biomedical research for its potent effects toward senescent cells, which are resistant to apoptosis and may be involved in physiological aging, as well as many pathological conditions [15]. As a hydrophobic agent, FIS readily penetrates cell membranes and accumulates in cells to exert neuroprotective, neurotrophic and antioxidant effects. FIS treatment may include alleviating inflammation, cell apoptosis and oxidative stress [16].

FIS inhibits the synthesis of pro-inflammatory and pro-oxidative cytokines in LPS-induced acute otitis media, alleviates cell apoptosis and inflammation in acute kidney injury, and may also alleviate LPS-induced oxidative stress and inflammation in neurodegenerative diseases [17].

Pharmacokinetics and metabolism studies of FIS in vivo have elucidated that geraldol (3,4′,7-trihydroxy-3′-methoxyflavone) is its active, methoxylated metabolite. It has been shown that it is the only one to locate in the nucleolus, and other structurally similar flavonols do not have such ability [18]. FIS is very hydrophobic in nature, resulting in limited water solubility (<0.8 mg/mL). This feature of FIS leads to poor systemic circulation, short half-life and poor cancer-targeted applications [19].

## 2. Spectrum of Fisetin’s Anticancer Activity

FIS induces apoptosis in various tumor cells by, for example, inhibiting cyclooxygenase-2, inhibiting the Wnt/EGFR/NF-κB pathway, activating the caspase-3 cascade, activating the caspase-3 and Ca^2+^ dependent endonuclease, and activating the caspase-8/caspase-3 dependent pathway via ERK1/2.

In addition, FIS controls the cell cycle and inhibits cyclin-dependent kinases (CDKs) in human cancer cell lines, modulates the pathways of lipid and protein kinases activated by AMP, and affects various signaling pathways, for example, by inhibition of PI3K/Akt/mTOR signaling [20], mitogen-activated protein kinases (MAPK) [21], and nuclear transcription factor (NF-κB) [22]. Reportedly, studies have shown that FIS inhibits aging by reducing p53, p21 and p16 expression in mouse and human tissues and macrophage-derived foam cells. There are also reports that FIS attenuates the pathogenesis of atherosclerosis in mice deficient in apolipoprotein E [23]. Moreover, FIS induces autophagic cell death by inhibiting both the mTORC1 and mTORC2 pathways [24].

### 2.1. Activity toward Bladder Cancer Cells

Bladder cancer is the sixth most common malignant tumor in men and the seventeenth in women [1]. Currently, intravesical infusions of BCG mycobacteria are the most effective and the most widely used therapeutic agent. Although the intravesical application of BCG is aimed at destroying the primary tumor, preventing its subsequent relapses and delaying them, side effects of BCG therapy are common and approximately one third of patients do not respond to the treatment [25]. Hence, it is necessary develop new effective agents that are generally better tolerated and cause fewer adverse reactions.

Few studies have been performed in bladder cancer models, although it appears that FIS may have promising results as an anticancer agent [26]. FIS significantly reduces the viability of T24, EJ and J82 bladder cancer cells in a time- and concentration-dependent manner [27]. In a rat model of bladder cancer induced by N-methyl-N-nitrosourea (MNU), FIS has been shown not to cause weight loss in the experimental animals, not to reduce food intake, and to show no apparent signs of toxicity, and in the FIS-treated group, bladder cancer is three times less frequent [28] and the mean mass of tumors in the FIS group is significantly lower than in the control groups [29].

FIS-induced apoptosis is modulated by two interrelated signaling pathways: p53 upregulation and NF-kappa B downregulation, resulting in a change in the pro- and anti-apoptotic protein ratio. Moreover, FIS inhibits cell proliferation by inducing apoptosis and blocking the cell cycle in the G0/G1 phase [27]. FIS significantly increases the expression of p53 and p21 proteins and lowers the levels of cyclin D1 [27,28], cyclin A, CDK4 and CDK2, thus contributing to cell-cycle arrest. FIS also increases Bax [27,28] and Bak [27] protein expression, but reduces the levels of Bcl-2 [27,28], Bcl-xL [27] and PCNA [28], and then starts the mitochondrial apoptotic pathway. These hypothetical pathways suggest that p53 activation and inhibition of the NF-kappa B system play an important role in FIS-induced apoptosis in bladder cancer cells [27]. Moreover, administration of FIS significantly reduces the expression of pIκB-α, IKKβ and NF-κB (p50 and p65) and increases the expression of p19ARF and IκBα [28].

### 2.2. Activity toward Breast Cancer Cells

Breast cancer is the most common cancer in the world and the most common among women. In 2020, 2.26 million new cases of breast cancer were detected and 0.68 million deaths were reported, making it the fifth leading cause of cancer death. Although this cancer occurs worldwide, its incidence is higher in developed countries [1,30]. Overexpression of the human epidermal growth factor receptor 2 (HER2) is observed in breast cancer. The main problem faced by medicine is the development of chemoresistance to HER2 inhibitors by cells in advanced stage of cancer [30].

Studies have shown that FIS induces cell apoptosis through various mechanisms such as receptor inactivation, induction of proteasome degradation, reduction in its half-life, reduction in enolase phosphorylation, and alteration of phosphatidylinositol 3-kinase/Akt signaling. FIS reduces HER2 tyrosine phosphorylation in a dose-dependent manner and aids in proteasomal degradation of HER2 rather than lysosomal degradation [31].

FIS inhibits the proliferation of 4T1, MCF-7 and MDA-MB-231 breast cancer cells in a concentration- and time-dependent manner [32], and, furthermore, it reduces cell migration and the ability to form a lineage by cells of the 4T1 and JC lines [32,33] and MDA-MB-231 and MDA-MB-468 [34]. The cytotoxic effect of FIS on MDA-MB-231 and MDA-MB-468 triple negative breast cancer (TNBC) cells has also been demonstrated. The cell growth inhibitory effect of FIS was also confirmed against ER-positive MCF-7 breast cancer cells and SK-BR-3 breast cancer cells overexpressing HER2. In addition, FIS has been shown to have little or no effect on normal cells at concentrations that are highly cytotoxic to breast cancer cells. The antiproliferative effect of FIS is associated with a significant reduction in the percentage of cells in the G1 phase of the cell cycle and a corresponding significant increase in the percentage of cells in G2/M [34].

FIS causes a significant reduction in histone H3 phosphorylation at serine 10 in breast cancer cells in the G2/M phase of the cell cycle, suggesting inhibition of Aurora B kinase in these cells. Treatment of FIS cells causes destabilization of the mitochondrial membrane and an increase in cytochrome c levels, which is consistent with the loss of mitochondrial membrane integrity. In addition, both caspase-8 and caspase-9 are cleaved, indicating activation of initiation caspases. PARP-1, which is involved in DNA repair and is one of the main targets for cleavage of effector caspases, is also cleaved in cells treated with FIS [34]. The cytotoxic effect of FIS is due to the induction of apoptosis. At lower concentrations (40 µM), FIS primarily induces early apoptosis, while FIS at higher concentrations (80 µM) primarily induces late apoptosis [32,34]. FIS causes a significant increase in the level of the Bax protein and a decrease in the level of the anti-apoptotic protein Bcl-x [32].

FIS reduces the enzymatic activity of both MMP-2 and MMP-9. Moreover, MMP-2, MMP-9 mRNA expression is also decreased. It induces the production of HO-1 in breast cancer cells and increases the expression of the HO-1 protein [33].

FIS enhances the activity of PI3K-3-phosphatidylinositol kinase and increases Akt phosphorylation; however, higher concentrations of FIS (25 or 50 µM) gradually reduce its phosphorylation [31]. The anticancer activity of FIS is also related to the regulation of the PI3K/Akt/mTOR pathway. Treatment of FIS cells significantly reduces the expression of Akt, P70 and mTOR. Moreover, p-IK and p-PI3K/PI3K, p-Akt and p-Akt/Akt, p-P70 and p-P70/P-70 and p-mTOR are significantly reduced [32]. FIS inhibits the activation of the PKCα/ROS/ERK1/2 and p38 MAPK signaling pathways. This effect is also associated with decreased activation of NF-κB [35]. It has been found that FIS may induce a new form of atypical apoptosis in MCF-7 breast cancer cells with caspase-3 deficiency, which is characterized by several features of apoptosis, including rupture of the cell membrane, mitochondrial depolarization, activation of caspase-7, -8 and -9, and cleavage of PARP. However, neither DNA fragmentation nor externalization of phosphotidylserine (PS) is observed in it. It has been reported that p53 protein is also activated by FIS and inhibition of autophagy by FIS seems an additional way to induce anticancer activity [36].

However, neither DNA fragmentation nor externalization of phosphotidylserine (PS) is observed in it. It has been reported that p53 protein is also activated by FIS and inhibition of autophagy by FIS seems an additional way to induce anticancer activity [32].

### 2.3. Activity toward Brain Cancer Cells

Although CNS tumors are rare, they are a significant cause of cancer morbidity and mortality, especially in children and young adults where they account for approximately 30% and 20% of cancer deaths, respectively [37]. FIS was confirmed to demonstrate antiproliferative activity against glioblastoma cells, and the IC50 value was 75 µM. The presence of apoptotic changes in cells treated with FIS has also been demonstrated, and this process is time- and dose-dependent [38]. Moreover, FIS effectively inhibits migration and invasion of glioblastoma cells [39]. FIS inhibits expression of the ADAM9 protein and mRNA, known to contribute to the progression of glioblastoma. FIS stably phosphorylates ERK1/2, which contributes to the inhibition of ADAM9 protein expression [39]. FIS increases the expression of caspase-3 and 9 and BAX protein, while reducing the expression of the anti-apoptotic protein BCL-2 and survivin [38]. Examination of the activation of the ERK1/2 and Akt signaling pathways found that FIS significantly induces the phosphorylation of ERK1/2 expression. In contrast, FIS did not significantly affect the phosphorylation of Akt expression, and, no significant effect on the total amount of ERK1/2 (t-ERK1/2) and Akt (t-Akt) proteins was observed, suggesting that the ERK1/2 pathway could be specifically activated in the pathway of cell migration and invasion inhibited by FIS [39].

### 2.4. Activity toward Cervix Uteri Cancer Cells

It is estimated that in 2020 there were 0.6 million new cases of cervical cancer and 0.34 million of deaths caused by this cancer. While the incidence and mortality of cervical cancer have been decreasing during the last several decade, this trend is mainly attributed to the widespread use of cervical screening [1,40]. However, the search for new chemical compounds that may inhibit the formation of this type of tumor is currently a priority in biomedical science.

Recent studies have shown that FIS reduces the viability of HeLa cervical cancer cells (IC50 = 36 ± 0.5 µM) [40] and inhibits cell invasion and migration in a concentration-dependent manner. Additionally, cells of other lines, including SiHa and CaSki treated with FIS, show a decrease in mobility [24]. Furthermore, FIS induces apoptosis of HeLa cells, activates caspase-3 and -8, and induces the cleavage of poly(ADP-ribose) polymerase, inducing the induction of apoptosis. However, the influence of FIS on caspase-9 activation has not been confirmed, thus far [41]. The combination of FIS and sorafenib synergistically induces apoptosis in HeLa cells, accompanied by a marked increase in the loss of mitochondrial membrane potential and activation of caspase-3 and caspase-8, which further increases the Bax/Bcl-2 ratio and causes PARP cleavage. In vivo studies using the HeLa xenograft model confirm that a combined treatment with FIS and sorafenib gives significantly better results than treatment with sorafenib alone or with FIS alone, and this synergism is based on the induction of apoptosis via loss of mitochondrial potential and the caspase-8/caspase-3 signaling pathway dependent on the DR5 death receptor [41].

Among the various elements of the plasminogen system, urokinase plasminogen activator (uPA), its receptor (uPAR) and plasminogen activator 1 and -2 (PAI-1 and PAI-2) play a major role in tumor progression and metastasis. Binding of uPA to uPAR is essential for the activation of plasminogen to plasmin, which in turn initiates a series of proteolytic cascades to degrade components of the extracellular matrix and thus causes migration of tumor cells from the primary site of origin to the distant secondary organ [42]. In tumor tissue concentrations, both PAI-1 and uPA promote tumor progression and metastasis [43]. The expression and activity of uPA is suppressed by FIS. The FIS-mediated alteration of uPA mRNA levels coincides with protein levels, suggesting that FIS may regulate uPA expression at transcription levels [24]. uPA is regulated by the MAPK or PI3K-Akt pathway in various types of tumors, including cervical cancer. Phosphorylation of p38 MAPK is selectively and strongly depressed without altering the amount of protein in FIS-treated cells, while the phosphorylation of ERK1/2, JNK1/2 and AKT show no significant changes, indicating that FIS inhibits uPA expression and reduces migration and invasion of cervical cancer cells via inactivation of the p38 MAPK pathway. Reportedly, FIS supports activation of ERK1/2 but does not support activation of p38 and JNK1/2, suggesting that ERK1/2 may be specifically activated in FIS-induced apoptosis. FIS-induced apoptosis is attenuated by inhibition of the activation of the ERK1/2 pathway with an inhibitor of the MEK1/2 pathway (PD98059), suggesting that this effect is specific for the ERK1/2 pathway [44]. Interestingly, the promoter activity of the *uPA* gene is suppressed by FIS, which disrupts NF-κB nuclear translocation and its binding amount on the *uPA* gene promoter [24].

### 2.5. Activity toward Colorectal Cancer Cells

According to GLOBOCAN 2020 data, colorectal cancer is the third most deadly and fourth most diagnosed cancer in the world; in 2020, nearly 2 million new cases and approximately 1 million deaths were detected. The incidence of colorectal cancer is steadily increasing worldwide, especially in developing countries [1,45]. Few drugs in the last decade have added to the bowel cancer treatment regimen and have significantly influenced prognosis. There are many challenges in expanding potential treatment options, in particular, better understanding the biology of the disease and the resistance mechanisms underlying cancer treatment failure. The development of new drugs is certainly one of the most important challenges in oncology [46].

FIS causes a decrease in the viability of colorectal cancer cells of the HCT116 and HT29 lines [47,48,49,50,51], LoVo [52], COLO205 [49,50,53], HT-29 [47,49,51,54] and HCT-15 [51]. It has also been shown that in the case of HCT-116 and HT-29 lines, FIS reduces the number of colonies, and this effect may be maximized by combining FIS with 5-fluorocuacil (5-FU) [47,50]. DNA integrity analysis confirmed that FIS induces apoptosis as manifested by the appearance of the DNA ladder [51,53]; additionally, FIS induces greater DNA fragmentation in HCT116 cells with zero securin and increases phosphotidylserine externalization. Moreover, FIS increases the cleavage of procaspase-3 and PARP in HCT116 cells, induces the p53 protein contributing to apoptosis. P53 phosphorylation has been shown to be induced by FIS in a time-dependent manner and correlated with the downregulation of securin expression, and higher levels of phosphorylated p53 are seen in securin-null cells treated with FIS. Thus, securin may inhibit FIS-induced apoptosis by antagonizing p53 activation [48]. FIS also works synergistically with N-acetyl-L-cysteine and the combination increases apoptosis in COLO205 colon cancer cells. Compared to treatment with FIS alone, combination therapy increases expression of cleaved caspase-3 and PAPR protein, and increases DNA ladder density. N-acetyl-L-cysteine reduces the mitochondrial membrane potential of FIS-treated cells, inducing caspase-9 cleavage [50].

FIS increases the cleavage of PARP, a key enzyme in the repair of DNA damage, and thus, cleaved PARP is generally considered a useful marker of apoptosis. FIS increases the cleavage of caspase-9, -3 and -7 in HCT-116 and HT-29 cells and reduces the number of cells with normal mitochondria and increases the number of cells with depolarized mitochondrial membranes. Additionally, FIS increases the levels of Smac/Diablo and cytochrome c in the cytoplasm and reduces the levels in the mitochondria [47]. FIS increases the cleavage of caspase-8 [49,52] and caspase-3 and the expression of cytochrome C. Surprisingly, treatment with FIS causes a higher level of cytochrome C release in chemotherapy-resistant cells than in nonresistant tumor cells [52]. In addition, the use of Hsp-9 protein inhibitors in combination with FIS causes a decrease in metalloproteinases (MMPs). However, this effect is not observed with FIS alone, nor is the induction of caspase-9 activity. In the presence of inhibitors of Hsp-90 and FIS proteins, both Bcl-XL and Bax proteins remain unchanged; however, the expression of Bcl-2 and p53 proteins is significantly decreased. Additionally, an increase in proteins targeting ubiquitin was detected in these cells [53]. Other studies have shown that treatment of FIS cells lowers the levels of the pro-survival proteins Bcl-2 and Bcl-XL, while increasing the levels of the pro-apoptotic Bak protein. FIS has not been shown to affect Bax levels [49,50], but Bax levels have been decreased in the cytoplasm and increased in the mitochondria of cells. FIS also increases the levels of Bak, BimEL, BimL and BimS proteins. No Truncated Bid was detected in the studies, while the Bid intact levels were decreasing, suggesting that FIS may induce Bid cut. Changes in the Bik protein level and an increase in Fas-L, TRAIL and DR5 levels were also confirmed in FIS-treated cells, but the Fas levels did not change [49].

The decline in cell numbers may be attributed to cell-cycle arrest. At 8 h after the addition of the FIS, the percentage of cells in the G1 phase increases with a concomitant decrease in the G2/M fraction. However, after 24 h, there is an accumulation of cells in the G2/M phase, with an equivalent decrease in the G1 phase [54]. FIS does not affect the level of CDK2 protein; it does not change the level of cyclin A, although the level of cyclin E is decreased. FIS increases p21 levels but has no effect on p27 levels. Treatment of HT-29 FIS cells resulted in a concentration-dependent increase in p21CIP1/WAF1 mRNA levels. The levels of CDK4 protein together with cyclin D1 and MMP-7 decrease after treatment with FIS [50,54].

FIS reduces the catalytic (p110) expression of the PI3K subunit and inhibits AKT phosphorylation in Ser473; however, inhibition of both proteins was more noticeable when cells were treated with the combination of FIS and 5-FU. Combination therapy leads to a greater decrease in mTOR and a remarkable increase in AMPKα within cells. AMP-activated kinase (AMPK) regulates the PI3K/AKT/mTOR pathway, and its activation causes cell-cycle arrest and tumor growth inhibition. The mTORC1 and mTORC2 protein levels in HCT116 and HT29 changed after treatment with FIS, although the effect was stronger with mTORC1 than with mTORC2. The amount of Raptor, Rictor and GβL, p-PRAS40, 4E-BP1, eIF4E and p70S6K is also significantly reduced. Thus, PI3K signaling is clearly influenced by FIS, and the effect seems to be additionally enhanced by 5-FU [46]. FIS also inhibits Wnt signaling activity by downregulating β-catenin and T4 cell factor, and inhibits the activation of EGFR and nuclear kappa factor B (NF-κB). It may induce apoptosis and inhibit cell growth by inhibiting Wnt/EGFR/NF-κB signaling pathways. Moreover, cells expressing COX2 are more sensitive to FIS [50]. Treatment with FIS significantly inhibits the levels of IGF1R and AKT phosphorylation in colon cancer cells [52].

In in vivo studies in mice, FIS has been shown to prevent tumor formation [47] and to inhibit tumor growth in a mouse xenograft model. Overall, the results gave rise to FIS as a promising agent for treating colon cancer [52].

### 2.6. Activity toward Kidney Cancer Cells

Although renal cell carcinoma (RCC) is responsible for approximately 2% of deaths worldwide, its incidence in developed countries has more than doubled over the last half-century and is today the 14th most common tumor worldwide [1,55]. Immunotherapy provides a modest but significant improvement in overall survival in metastatic renal cancer. Current progress in molecular biology recognizes many pathways involved in the progression of this tumor. Several strategies targeting these pathways have been explored with major clinical benefits demonstrated in early studies [56]. FIS has been found to decrease the viability of 786-O, A-498, ACHN [57] and Caki-1 [57,58], and to reduce colony formation by these cells in a dose-dependent manner and to cause the stop of cell cycle in G2/M phase. FIS upregulates p21 and p27 and lowers cyclin B1 levels. FIS inhibits the migration and invasion of renal cancer cells by suppressing CTSB, CTSS and ADAM9 [57], and increases ERK phosphorylation without significant differences in p38 or JNK phosphorylation.

FIS also induces a sub-G1 fraction of the cell cycle. The cleavage of poly(ADP-ribose) polymerase (PARP) is increased, which is manifested, among others, by altering cell morphology, causing cells to contract and form vesicles. In cells treated with FIS, chromosomal damage and DNA fragmentation occur, confirming that FIS induces apoptosis in tumor cells. FIS significantly increases caspase activation, and expression levels of Fas, c-FLIP, FADD, Bcl-2, Bcl-xL and PUMA are not altered after treatment with FIS. However, it induces upregulation of DR4 and DR5 death receptor expression in a dose-dependent manner. In addition, FIS modulates DR5 expression at the transcriptional level, induces expression of ER stress-related proteins including CHOP and the activating transcription factor (ATF4), increases the multiplication form of X-box binding protein (XBP)-1 mRNA, and clearly induces p53 protein expression. However, FIS did not affect p53 mRNA levels. Research indicates that it increases p53 expression by inducing protein stability. Moreover, ROS signaling has been shown not to be involved in FIS-induced apoptosis [58].

### 2.7. Activity toward Leukemia Cancer Cells

Leukemia, a malignant tumor of the blood characterized by transformed hematopoietic progenitors and diffuse bone marrow infiltration, was the 10th leading cause of cancer death in the world in 2020. Leukemia was responsible for approximately 3.4% of all new cancer cases and 3.8% of all cancer deaths in 2020 according to the Surveillance, Epidemiology, and End Results (SEER) Program [1,59,60]. With the advent of new therapies such as mutation-targeting inhibitors, pro-apoptotic agents, T-cell therapy and immunotherapy, leukemia mortality has decreased, although it is still a very widespread disease [59]. The current research focuses on identifying new chemical and biological substances susceptible to specific inhibitors, designing the best strategies for combining these new agents with traditional chemotherapy regimens [61].

FIS is deemed not highly toxic to the cells of the myeloid leukemia K562 line [62,63] and its stronger activity was confirmed on WEHI-3 [64], HL-60 [65,66], THP-1 and U937 [67]. At low concentrations (10 and 20 µM), FIS does not cause visible changes in the viability of K562 cells; however, at higher concentrations (30–50 µM) it leads to a statistically significant, though slight, decrease in cell survival [62]. The results showed that FIS exhibited different sensitivity to leukemia cells, inhibiting their growth in a dose-, time- and cell-line-dependent manner. IC50 values after 48 and 72 h were found to be high for K562 cells of 163 and 120 µM, respectively [63], while for HL-60 cells the IC 50 significantly decreased to 82 and 45 µM.

FIS also increases apoptosis in HL-60 cells [65], but has not been shown to be a potent inducer of apoptosis in K562 cells. At a concentration of 10 µM, the increase in the number of apoptotic cells is significant only after longer exposure times; however, at a concentration of 20 µM, the increase in apoptosis is significant [62]. FIS, on the other hand, is very effective in inducing apoptosis in HL60 cells. Loss of myotochondrial membrane potential is an important sign of apoptosis because it is associated with the initiation and activation of apoptotic cascades. The treatment of HL60 FIS cells causes just an increase in the loss of mitochondrial membrane potential, disruption of the mitochondrial membrane potential along with increased caspase-3 activity [65]. FIS induces an increase in intracellular Ca^2+^ but reduces the production of ROS in WEHI-3 cells [64]. The cytotoxic effect of FIS is accompanied by the appearance of apoptotic features, including DNA fragmentation, apoptotic bodies, and an increase in the sub-G1 ratio. FIS treatment results in a rapid and transient induction of caspase-3/CPP32 activity without affecting caspase-1 activity. Furthermore, PARP cleavage and a decrease in procaspase-3 are present in FIS-treated cells. FIS causes an increase in the pro-apoptotic protein bax and a decrease in the anti-apoptotic protein Mcl-1. However, all Bcl-2, Bcl-XL and Bad proteins remain unchanged in HL-60 cells [66]. FIS modulates the level of transcription of genes associated with apoptosis. It increases the expression of *BAX* and *BCL2*, resulting in the ratio of *BCL2* to *BAX* being higher than 1. It also increases the levels of caspase-3 and AIF mRNA, but also increases necrosis markers including RIP3 and PARP1 [62]. FIS activates the caspase-8 initiator and cleaves the receptor-mediated Bid protein, which is the receptor-mediated junction between caspase-8, and caspase-9 from the mitochondrial pathway. Executive caspases-3 and -7 are also split. The cleavage of PARP confirmed the functional result of the cleavage of caspases-3 and -7 as it is a substrate for caspase-3 and -7. Activation of the caspase-9 initiator was also seen [62,66]. FIS increases the activity of caspase-3, -8 and -9. Cells pretreated with caspase inhibitors and then treated with FIS show an increased number of viable cells. FIS reduces the expression of cdc25a, but increases the expression of p-p53, Chk1, p21 and p27, which may lead to a G0/G1 arrest. However, other studies indicate that FIS inhibits the anti-apoptotic proteins Bcl-2 and Bcl-xL and increases the pro-apoptotic proteins Bax and Bak. Moreover, it increases the expression of cytochrome c and AIF [64]. In addition, FIS causes the arrest of the S and G2/M cell cycle [63]. Treatment of HL60 cells with FIS results in a dose-dependent increase in the percentage of cells in the G2/M phase, accompanied by a decrease in the percentage of cells in the G0/G1 phase [67].

Interestingly, in some cases, FIS-treated K562 cells show a much greater ability to invade through the matrix and a greater ability to migrate across membranes compared to control cells [62]. FIS causes an increase in F-actin staining and its more visible redistribution toward the plasma membrane. Moreover, FIS enhances the nuclear localization of β-catenin and influences the expression of genes related to metastasis by increasing the levels of *CD44, MMP2*, *SNAIL*, *SLUG*, *VIM* and *ROCK1* mRNA, but has no effect on *RHOA* expression. In addition, significant upregulation of *MMP9*, *TWIST*, *PYK2* and *CTNNB1* expression is observed. Additionally, it has been shown that very low concentrations of FIS also increase the level of MMP9, TWIST and PYK2 mRNA [61]. FIS also causes upregulation of genes such as: nuclear factor kappa-b inhibitor alpha (*NFKBIA*), growth arrest and DNA damage-inducible alfa (*GADD45A*), cyclin-dependent kinase inhibitor 1A (*CDKN1A*), phorbol-12-myristate-13-acetate-induced protein 1 (*NOXA*) and NF-kappa-B inhibitor zeta (*NFKBIZ*). The concentration-dependent effect of the FIS used was shown by the genes’ growth arrest and DNA damage-inducible beta (*GADD45B*), cyclin dependent kinase inhibitor 2D (*CDKN2D*), Thioredoxin Interacting Protein (*TXNIP*) and Mothers Against DPP Homolog 5 (*SMAD5*). Under the influence of FIS, some genes were downregulated, i.e., *MYC*, *MYB*, *C-KIT*, *TUBA1A* and *TUBAL3*, in addition, Forkhead box K1 (*FOXK1*), Forkhead box A2 (*FOXA2*), (*BCL-XL*) and ATP-binding cassette transporter (*ABC*) [63].

In addition, FIS affects tissue factor pathway TFPI inhibitor, inhibitor of DNA binding-1 and -3, heat shock protein family a (hsp70) member 1b (HSPA1B) and isocitrate dehydrogenase (NADP (+)) 1 (IDH1). The upstream activation of the mitogen-activated protein kinase kinase 1 (*MAP3K1*), caspase-4, sphingoid base N-palmitoyltransferase (*CERS6*), E3 ubiquitin-protein ligase CBL-B (*CBLB*) genes, while mitochondrial Lon protease (*LONP1*), signal transducer and activator of transcription 3 and 5A (*STAT5A* and *STAT3*) and Janus Kinase 1 (*JAK1*) were some of the downregulated genes in FIS-treated cells [65,67].

Studies have also confirmed several important signaling pathways that are altered by FIS, including JAK/STAT pathway, KIT receptor signaling and growth hormone receptor signaling [63], and mitogen-activated protein kinases (MAPK) [67]. FIS significantly modulates the expression of genes involved in cell proliferation and division, apoptosis, cell-cycle regulation and other important cellular processes, such as replication, transcription and translation [63]. Moreover, FIS inhibits elements of the mTORC1 pathway by lowering p70 S6 kinase levels and inducing hypophosphorylation of S6 RiP kinase, eIF4B and eEF2K [66], and overexpresses the autophagy gene, *BECN1*. However, its influence on other genes involved in the process of autophagy, such as *LC3B* and *p62*, has not been confirmed. The mRNA level of the major negative regulator of autophagy, mTOR, increases after treatment with FIS. The expression of two other critical nodes in the PI3K/AKT and mTOR signaling pathways promoting survival is also dysregulated [62]. Moreover, in vivo studies have shown that FIS is able to kill clothed cells, causing the tumor to shrink in a mouse xenograft model [67].

### 2.8. Activity toward Liver Cancer Cells

Liver cancer is a major contributor to the global cancer burden, and incidence rates have increased in many countries in recent decades. As the major histological type of liver cancer, hepatocellular carcinoma (HCC) is responsible for the vast majority of diagnoses and deaths [68]. Primary liver cancer is the sixth most common cancer in the world and the third most common cause of cancer mortality [1]. Whilst new treatment options are needed for patients with liver cancer, the use of natural compounds and/or nanotechnology may provide patients with better outcomes with less systemic toxicity and fewer side effects. Improved treatment may lead to better prognosis [69].

Studies have shown that FIS inhibits the proliferation of HCC-LM3, SMMC-7721 [70], HepG2 [71,72] and SK-Hep-1 [73] liver cancer cells. FIS influences invasion and migration of liver cancer cells by markedly reducing them, and concomitant treatment with FIS and bromocriptine further reduces the number of affected cells, suggesting that FIS may have a similar molecular mechanism to bromocriptine in regulating liver cancer progression. FIS significantly reduces the number of colonies of tumor cells [69], induces an increase in the percentage of cells in the S phase [72].

Epithelial–mesenchymal transition plays a key role in cancer metastasis, and TGF-β1 induces the mesenchymal–epithelial transition process. Studies have shown that FIS reduces TGF-β1 by inhibiting this process. Moreover, vimentin, E-cadherin and N-cadherin are clearly reduced under the influence of FIS, which is consistent with the expression of TGF-β1. FIS may ameliorate liver cancer progression by inhibiting EMT through the TGF-β1 signaling pathway.

Moreover, FIS induces an increase in the number of apoptotic and necroptotic cells [72] and in combination with bromocriptine significantly increases the level of apoptotic liver cancer cells [70]. FIS-treated cells show distinct features including cell shrinkage and reduced confluence. HepG2 cells exposed to FIS show marked apoptotic features including chromatin condensation, cell swelling and nuclear fragmentation, which become more visible with increasing concentration. Chromatin condensation and nuclear fragmentation are important indicators of apoptosis. In addition, apoptotic cell surface morphology is observed, i.e., formation of cell membrane vesicles and separation of apoptotic bodies [72,73]. It is known that the apoptotic cascade is initiated by reduction in the mitochondrial membrane potential Δψm. In HepG2 cells treated with FIS, a collapse in the potential Δψm will be observed. Additionally, FIS increases the expression of Bax and causes a decrease in the expression of the anti-apoptotic protein Bcl2. This disproportion in the expression of pro- and anti-apoptotic proteins stimulates the process of apoptosis [72]. FIS also induces the cleavage of caspase-3. Importantly, the high concentration of FIS results in a clearly more intense cleavage of caspase-3 and PARP. These results show that activation of the caspase signaling pathway is involved in FIS-induced apoptosis, which may be a possible molecular mechanism by which FIS exhibits potent anticancer activity [70]. FIS does not induce caspase-1 activity. Moreover, the caspase-3 inhibitor Ac-DEVD-CHO, but not the caspase-1 inhibitor Ac-YVAD-CHO, reverses the cytotoxic effect of FIS. Moreover, in the cells treated with FIS, not only cleavage of the caspase-3 substrate-PARP but also the D4-GDI protein and a decrease in the procaspase-3 protein were detected [73].

FIS does not affect CDK2 protein levels and change the cyclin A level, whereas that of cyclin E is decreased. FIS increases the p21 levels; however, the p27 level is not affected by FIS treatment. FIS decreases cyclin E levels and increases p21 levels in HT-29 cells. Furthermore, FIS decreases CDK2 and CDK4 activity in cells and FIS decreases CDC2 activity (Figure 2). FIS reduces autophagy flux formation. The mRNA of the *mTOR*, *Atg5*, *Atg16L* and *LC3A* genes have been shown to be upregulated, while the mRNA levels of the *Atg7* and *Beclin1* genes are downregulated. FIS inhibits the expression of *Atg7*, *Atg16L*, *mTOR* and *pACC*, and increases the expression of *Atg5*, *AMPKα*, *AMPKβ1/2*, *ACC* and *Akt*, confirming that FIS inhibits autophagy by activating PI3K/Akt/mTOR and modulating AMPK signaling [71]. FIS has also been shown to significantly reduce the expression of VEGFR1, p-ERK1/2, p38 and Pjnk.

In vivo studies found that FIS significantly reduced the weight of tumors isolated from mice with orthotopically implanted tumors and it significantly prolongs the survival time of mice with tumors. Dopamine was found to be highly expressed in serum and tumor tissue, and the use of FIS significantly lowers it. Administration of FIS also significantly lowers the level of the cell proliferation marker Ki-67. Additionally, TGF-β1 is reduced in tumor tissue samples isolated from FIS-treated mice, confirming that FIS may inhibit the development of liver cancer [70].

### 2.9. Activity toward Lung Cancer Cells

In the last century, lung cancer has evolved from a rare and obscure disease to the second most common cancer in the world and the leading cause of cancer deaths. An estimated 2.2 million new cases and 1.8 million deaths were diagnosed worldwide in 2020. Incidence trends and geographic patterns differ between men and women and primarily reflect historical, cultural and regional differences in smoking. Known risk factors for lung cancer include behavioral, environmental and genetic risk factors, all of which play a role in tumor development and also affect individual patient response ability [1,74]. Despite progress in treatment, the prognosis remains unsatisfactory due to late diagnosis. Many genomic studies have linked lung cancer with genomic changes that affect many cell functions, including cell growth, differentiation, proliferation, survival, mobility and invasion.

It has been proven that flavonols exhibit antiproliferative and pro-apoptotic effects, inhibiting invasion and metastasis [75]. FIS effectively inhibits the growth of A549 and NCI-H460 lung cancer cells [76,77,78,79]. FIS in concentrations from 0 to 10 μM does not affect cell viability; however, its use at concentrations of 20–40 μM significantly reduces the viability of lung cancer cells [79].

FIS influences proliferation related genes such as *cyclin D1*, *c-myc* and cyclooxygenase *(COX)-2* by downregulating them. It causes the accumulation of cells in the G2/M phase and the reduction in the G1 phase of the cell cycle, and increases the mRNA expression levels of genes related to the cell cycle: cyclin D kinase inhibitor *(CDKN) 1A*, *CDKN1B* and *CDKN2D* [76]. It has been shown that invasiveness and motility of FIS-treated cells is suppressed. This is confirmed by the mRNA expression of matrix metalloproteinases MMP-2 and -9, which are also downregulated by FIS. It is therefore suggested that FIS inhibits the adhesion, invasion and migration of lung cancer cells by regulating the expression of related genes [76,79]. FIS exhibits a concentration-dependent inhibitory effect on the cell adhesion capacity [79] and in combination with paclitaxel it causes disorganization of the actin cytoskeleton and almost complete loss of actin stress fibers. Moreover, simultaneous treatment with FIS and PTX causes a partial degradation of the actin cytoskeleton in a large proportion of A549 cells, an effect that is manifested by a significant reduction in F-actin fluorescence [77]. FIS leads to a significant reduction in MMP-2 and u-PA at the protein level and inhibits their activity. FIS also has an inhibitory effect on MMP-2 and u-PA at the mRNA level. These data suggest that FIS may regulate the expression of MMP-2 and u-PA at the transcriptional level [79].

FIS in combination with PTX significantly reduces the expression of CDH2, FN1, MMP-2, PLAU, SNAI2 and TWIST, and significantly increases the level of CDH1 and TJP1 transcript and less markedly increases the OCLN mRNA, but has a negligible inhibitory effect on the expression of MMP-9, SNAI1 and VIM. Simultaneous treatment with FIS and PTX had an inhibitory effect on mRNA expression of the *PI3K*, *AKT* and *mTOR* genes. The antimigratory and anti-invasive effects of the combination of FIS and PTX may be associated with inhibition of the PI3K/AKT/mTOR signaling pathway [77]. FIS also lowers p-ERK1/2 levels, ERK1 and ERK2 activation and MEK1 activity, among others, by reducing the phosphorylation of ERK1 and ERK2. Since p-MEK1 is downregulated by FIS, this suggests that it inhibits MEK1/2 activation of the ERK signaling pathway to reduce cell growth and invasion [76,79]. FIS did not significantly affect the activity of phospho-JNK1/2, phospho-p38 and Akt. Moreover, it does not change the total amount of ERK1/2, JNK1/2, p38 and Akt proteins [79].

FIS also induces apoptosis in lung cancer cells. The levels of histone-associated cytoplasmic DNA fragmentation, the formation of apoptotic bodies, and the number of cells in the sub-G1 phase are noticeably higher in FIS-treated cells. Additionally, it increases the expression of the Bax protein and reduces the expression of the Bcl-2 protein, and significantly increases the expression of active (cleaved) caspase-9 and caspase-3 [76,78]. The mRNA expression levels of anti-apoptotic genes (*Bcl-2*, *CXC 4*, *CD44* and *E-cadherin*) are also inhibited, which is consistent with an increase in the level of apoptosis [76].

### 2.10. Activity toward Melanoma Cancer Cells

Melanoma is a potentially fatal tumor that occurs most frequently on the skin, but may also arise in mucous membranes and elsewhere (e.g., the eyeball). The worldwide incidence of melanoma has increased rapidly over the past 50 years. Malignant melanoma is the most lethal form of skin cancer that tends to metastasize beyond its original site. When melanoma is advanced, surgery is no longer sufficient and the disease becomes more difficult to treat [80]. It is estimated that in 2020, melanoma accounts for 325,000 new cases, which is 1.7% of global cancer diagnoses according to GLOBOCAN. The standardized incidence rate is 3.8/100,000 for men and 3/100,000 for women, with a cumulative risk of life of 0.42% and 0.33%, respectively [1,81].

Treatment options for metastatic melanoma have expanded in recent years and are directly dependent on the stage of the disease at diagnosis and the extent of metastasis. The therapy used includes several drugs with different mechanisms of action, including chemotherapy, immunomodulating agents, BRAF serine/threonine protein kinase and mitogen-activated protein kinase (MEK) inhibitors. Hence, it is important to search for substances with antimelanoma activities [82].

Studies have shown that FIS affects melanoma cells: M17, SP6.5 [83] A375, 451Lu [84,85], A375, SK-MEL-28 and RPMI-7951[86]. The viability of uveal blackening cells decreases depending on the dose of FIS applied. The IC50 value is 59 µM for M17 cells and 84 µM for SP6.5 [83]. Treatment with FIS of 451Lu cells at various time points showed a dose-dependent reduction in cell viability. The IC50 was estimated to be 80, 37.2 and 17.5 µM after 24, 48 and 72 h, respectively (Table 1). FIS (5–20 µM) reduces the invasive potential of melanoma cells, and studies using the 3D equivalents of melanoma, consisting of A375 cells mixed with normal human keratinocytes embedded in a fibroblast matrix, confirmed that treatment with FIS also reduced the size of the infiltrates and invasive characterization of A375 cells [85].

FIS stops cells in the G1 phase of the cycle, and, inhibition of the cell cycle correlates with a reduction in the level of cyclin-dependent kinases (cdk)-2, -4 and -6, key cycle regulatory proteins involved in G1 phase progression. FIS also changes the localization of β-catenin. In untreated cells, β-catenin is mainly located in the cytosolic compartment; however, with increasing FIS doses, the cytosolic fraction of β-catenin decreases with a simultaneous decrease in nuclear β-catenin, which indicates that a significant amount of β-catenin is phosphorylated and degraded, resulting in reduced accumulation in the testes [85].

Interestingly, BRAF mutant melanoma cells are more sensitive to FIS, which is associated with a reduction in MEK1/2 and ERK1/2 phosphorylation. FIS inhibits IKK activation, leading to a reduction in activation of the NFκB signaling pathway and promotes the mesenchymal–epithelial transition in melanoma cells, which is associated with a decrease in mesenchymal markers (N-cadherin, vimentin, cochlea and fibronectin) and an increase in epithelial markers (E-cadherin and desmoglein) [86]. FIS also causes a dose-dependent decrease in the expression of the growth factor Wnt and Wnt5a. Interestingly, the decrease in coreceptor (FZD/LRP-6) expression along with the decrease in DVL expression coincides with the increase in endogenous Wnt inhibitors (DKK1 and WIF) in FIS-treated cells [85]. Thus, FIS has a functional role in Wnt/β-catenin signaling in the context of inhibiting cell growth.

FIS has also been shown to be effective in inducing apoptosis, which differed depending on the cell type, e.g., A375 cells are more susceptible to FIS than 451Lu cells [84]. The intrinsic pathway of mitochondrial-mediated apoptosis is regulated by the Bcl-2 family of proteins. FIS significantly increases the expression of Bad, Bax and Bak (pro-apoptotic protein) levels [83] and reduces the expression of Bcl-2 [84], Mcl-1, Bag-1 and Bcl-xL (anti-apoptotic protein). Additionally, FIS increases the permeability of mitochondria and increases the levels of cytochrome c in the cytoplasm [83]. FIS increases the levels of activated caspase-8 [84], -9 and -3 [83] and induces the expression of TNFR-1 and TNFR-2 death receptors, but has no effect on DR-3, -4 and -5, which in association with the protein Bcl-2 and Bax indicate concomitant activation of both external and internal apoptotic pathways [84].

### 2.11. Activity toward Stomach Cancer Cells

Stomach cancer remains one of the most common (fifth) and most deadly (fourth) tumors worldwide, especially among older men. The incidence and mortality of stomach cancer varies greatly and is strongly dependent on diet and Helicobacter pylori infection. While progress in the prevention and treatment of *H. pylori* infections has reduced the overall incidence of stomach cancer, it has also contributed to the increased incidence of cancer of gastric cardia, a rare cancer subtype that has grown sevenfold in recent decades [118]. The American Cancer Society estimates that in the United States in 2022, there were approximately 26,380 new cases of stomach cancer and approximately 11,090 deaths due to this cancer. In 2020, over 1 million new cases and almost 0.77 million deaths were diagnosed worldwide [1].

FIS inhibits the proliferation of stomach cancer cells and reduces the rate of proliferation in all stomach cancer cell lines tested, i.e., SGC7901, GES-1, AGS and SNU-1 [106,107]. Moreover, FIS increases the percentage of cells in the G2/M phase while reducing cells in the S phase [106]. It has also been shown that cells grown without the addition of serum for 24 h to induce a cycle arrest in the G0/G1 phase and then cultured in serum medium enter the S phase relatively quickly (after 9 h), and the addition of FIS causes the cells to remain in the G0/G1 phase; in addition, FIS causes cells to remain in the G1 phase even after 48 h [107].

FIS increases the level of the cyclin-dependent kinase inhibitor (CDKI) Cip1/p21 in cells at lower concentrations (25–50 µM), and lowers it at higher concentrations (75 µM). With Kip1/p27, there is an initial increase in level followed by a decrease at higher concentrations. The Cip/Kip proteins are upregulated by several pathways, both at the transcriptional and post-transcriptional levels. It has also been shown to increase total p53 levels in a time- and dose-dependent manner. At the same time, a decrease in the expression of Mdm2, the negative regulator of p53, is observed. FIS reduces the expression of the regulatory protein levels of cyclin G1 and CDK, including CDK2, CDK4, cyclin D1 and cyclin. The results suggest that FIS induces G1 arrest in stomach cancer cells by modulating the expression levels of CDK-cyclin-CDKI and p53 [107]. An inhibitory effect of FIS on ERK 1/2 activation has also been demonstrated [106].

FIS also causes the dissociation of JC-1 from dimer form to monomer form. The monomer/dimer ratio increases in the presence of FIS, suggesting a role of mitochondrial-mediated apoptosis as one of the mechanisms of cell death [107]. FIS induces the process of apoptosis already at a concentration of 15 µM. The percentage of apoptotic cells increases to about 90% [106]. In addition, there is a clear increase in the level of caspase-7 expression after treatment with FIS cells. The expression level of procaspase-7 remains slightly decreased. There is also a marked reduction in the expression of the anti-apoptotic proteins Bcl-2 and Bcl-x. Bim expression increases and Bid decreases. The use of FIS also significantly reduces the activation of ERK 1/2 [106]. FIS also causes an increase in the cleavage of caspase-3 and PARP and an increase in the level of Bax [107].

### 2.12. Activity toward Ovary Cancer Cells

Ovarian cancer is the eighth most commonly diagnosed tumors among women in the world. Epithelial ovarian cancer is the most dominant pathological subtype, with five major histotypes that differ in origin, pathogenesis, molecular alteration, risk factors and prognosis. Epidemiological data indicate that in 2020 over 0.3 million ovarian cancers were detected and 0.2 million deaths were reported among women [119]. Recently approved therapies have made significant progress in the treatment of ovarian cancer; however, more treatment options are needed to further improve outcomes in patients with advanced disease [120]. Cisplatin chemotherapy usually has few side effects, and cellular resistance to cisplatin is common. In order to overcome these problems, the use of combination therapies with the use of natural substances was considered.

Combined treatment with FIS and cisplatin effectively inhibits the proliferation of A2780 ovarian cancer cells and enhances the apoptotic effect. Changes in gene expression also indicate that in the treated cells, the pro-apoptotic expression of *Bax* genes increases and the anti-apoptotic expression of *BCL-2* genes decreases, moreover, the expression of caspases 3 and 9 genes also increases significantly [109]. The inhibition of anti-apoptotic proteins is a key goal in stimulating the apoptotic process in tumor cells. It is therefore essential to determine the interaction of FIS binding with potential anti-apoptotic target proteins (Bcl-2 and Bcl-xl). FIS shows better binding affinity for Bcl-xl than for Bcl-2. In the docked FIS-Bcl-xl complex there are a total of four hydrogen bonds, four hydrophobic contacts and one electrostatic interaction, which explains its high binding affinity for Bcl-xl [121].

### 2.13. Activity toward Pancreas Cancer Cells

Pancreatic cancer is the seventh leading cause of cancer death worldwide and deaths from it are higher in more developed countries. As patients rarely show symptoms in the initial stage of the disease, it remains one of the most lethal malignant tumors, causing 0.47 million deaths in 2020. Worldwide, nearly 0.5 million new cases of pancreatic cancer were reported in 2020, despite progress in the detection and treatment of pancreatic cancer, the 5-year survival rate is still only 9% [1,122]. Pancreatic cancer still has a very poor prognosis, even in a relatively small group of patients (maximum 15%) who have been diagnosed with a clearly resectable tumor [123]. Currently, research is focused on progress provided by novel potential targets to effectively extend life expectancy [124].

Pancreatic adenocarcinoma is an exceptionally malignant tumor associated with low survival rates. Studies have confirmed that FIS reduces the viability of pancreatic cancer cells in four cell lines (PANC-1, BxPC-3, MiaPACA-2 and HPC-Y5). Interestingly, low concentrations of FIS (25.50 µM) do not effectively inhibit the viability of HPC-Y5 and PANC-1 cells, while BxPC-3 and MiaPACA-2 cells are sensitive to a low concentration of FIS [111,112]. Thus, the differential sensitivity may be tumor type specific [112]. Treatment of AsPC-1 cells (multidrug resistant line) with FIS resulted in growth inhibition and the IC50 value was 38 µM. The AsPC-1 cell line, highly resistant to available chemotherapeutic agents, shows extraordinary sensitivity to FIS treatment. In addition, FIS significantly reduces thymidine incorporation [113], induces cell-cycle S-phase arrest and inhibits G1 phase, and levels of gamma-H2AX foci are markedly increased in FIS-treated cells. Moreover, levels of RFXAP mRNA and other DNA repair genes such as *RAD50*, *BRCA2*, *RPA2* and *POLD3* are activated by the action of FIS [123]. Studies have shown that genes mainly related to cell-cycle control, invasion and metastasis (*p21*, *p16*, *IκBα*, *NME4*, *KISS1*) are activated, and the downregulated genes reported to be anti-apoptotic, among others, *BCL2L1* and genes involved in cell invasion, proliferation and metastasis such as *NF-κB*, *MMP9*, *EGFR* and *HER-2*. FIS inhibits pancreatic cancer by clearly targeting the PI3K/AKT/mTOR signaling cascade rather than the JAK2 cascade. FIS significantly reduces the migration and invasion capacity of pancreatic cancer cells. FIS suppresses the expression of MMP-2 and MMP-9, which are involved in cellular metastasis. The data illustrate that FIS may reduce the infiltration capacity with migration. The expression of AKT, JAK2 and p-JAK2 proteins in the FIS-treated groups shows no significant changes, while p-AKT is decreased. Moreover, total mTOR is not changed; nevertheless, p-mTOR and PI3K are significantly decreased, indicating that the PI3K/AKT/mTOR cascade is involved in the inhibition of FIS-treated cells. This suggests that FIS may inhibit growth, invasion and migration by suppressing the PI3K/AKT/mTOR cascade [5,111]. FIS also causes IK-Kα/β inhibition, phosphorylation and degradation of IκBα, and subsequent activation of NF-κB [111].

FIS does not reduce the content of vimentin; however, it significantly increases the content of the epithelial marker E-cadherin and weakens the content of the interstitial markers N-cadherin and a-SMA, which results in a reduction in the extracellular matrix (ECM) secreted by stromal tumor cells [5].

FIS activates apoptosis [123] and causes a strong increase in caspase-3 and -8 activation and an increase in PARP cleavage, indicating that it promotes cell apoptosis through a mitochondrial-dependent cascade [5,111]. In addition, the death domain containing TNFR1 and TNFRSF25 (DR3) receptors shows a strong modulation after FIS treatment.

It was noted that FIS induces autophagy by increasing the autophagy marker LC3B, which results in the appearance of autophagous vesicles. The AMPK/mTOR pathway is well established as a critical regulator of this process. The levels of p-AMPK increase in response to the FIS, while the level of p-p70s6k decreases. In addition, FIS increases the level of Parkin and PINK1. Interestingly, PERK, ATF4 and ATF6 expression is also upregulated by FIS. Moreover, the expression of LC3B, Parkin, ATF4 and ATF6 in vivo is also increased [112].

### 2.14. Activity toward Prostate Cancer Cells

Prostate cancer is the second most common cancer diagnosis in men and the fifth most common cause of death in men worldwide. It may be asymptomatic at an early stage, and is often slow, requiring only active supervision. In 2020, over 1.4 million new cases of prostate cancer were reported worldwide, with a higher incidence in developed countries, and approximately 0.4 million deaths. Although there is no evidence yet on how to prevent prostate cancer, it is possible to reduce the risk by limiting high-fat foods and increasing the consumption of vegetables and fruits [1,125]. Concerning prostate cancer, several epidemiological studies have shown an inverse relationship between the consumption of flavonols and its incidence [126].

FIS causes dose- and time-dependent loss of viability of prostate cancer cells at concentrations above 20 µM [117]. In addition, it was shown that FIS in combination with car-bazitaxel reduces cell viability and significantly inhibits colony formation. FIS has been shown to synergistically and significantly increase the sensitivity of prostate cancer cells to cabazitaxel treatment and inhibit cell viability and long-term clonogenic growth of prostate cancer cells [114]. Arguably, FIS exhibits a dose-dependent inhibitory effect on the adhesion and migration capacity of prostate cancer cells. It is correlated with a decrease in the level of protein, mRNA and the activity of MMP-2 and MMP-9 [125].

It has also been observed that some cells are less sensitive to FIS due to the presence of the PTEN-null mutation, which causes constitutive activation of kinase Akt, which is correlated with the progression of prostate cancer. FIS is capable of inhibiting mTOR activity, mTOR autophosphorylation in S2481 and Akt-mediated phosphorylation in S2448. Raptor and Rictor, mTORC1 and mTORC2 companion proteins, respectively, are also lowered by FIS. The level of total and phosphorylated proline-rich PRAS40 Akt substrate, which binds Raptor to mTORC1 formation, is significantly lowered after FIS treatment. This indicates that FIS inhibits PRAS40 phosphorylation, leading to inhibition of substrate binding to mTORC1. Finally, the level of G protein of the β-like protein (GβL), which is part of both mTORCs, is significantly lowered by treatment with FIS. Akt has been shown to be elevated in prostate cancer and its expression correlates with cancer progression. Act is the direct target of mTORC2 and its phosphorylation at S473 by mTORC2 contributes to its full activation. FIS is effective in reducing Akt phosphorylation in both T308 and S473. Treatment of FIS cells lowers p-Akt levels. In addition, FIS dose-dependently increases AMPK phosphorylation. These results indicate that it inhibits mTOR by inactivating Akt and activating AMPK, resulting in the activation of the TSC complex [116]. FIS also inhibits the transcriptional activity of NF-κB and AP-1. In particular, the binding activity of NF-κB and AP-1 is strongly inhibited by FIS at a concentration of 20 µM. FIS-treated cells also show a decrease in NF-κB, c-Fos and c-Jun protein levels [117].

FIS has also been shown to act synergistically with TRAIL to induce apoptosis. The percentage of apoptotic cells was clearly increased due to the sensitization of cells resistant to TRAIL-dependent apoptosis. Moreover, FIS significantly increases the expression of TRAIL-R1 but does not affect the expression of TRAIL-R2. Stimulation of death receptors induces the formation of DISCs, which results in the recruitment and activation of caspase-8. TRAIL and FIS alone activate caspase-8; however, the simultaneous incubation of cells with TRAIL and FIS significantly increases the activity of caspase-8 and -3. Additionally, TRAIL with FIS significantly increases the loss of ΔΨm potential and reduces the activity of NF-κB. FIS blocks TRAIL-induced NF-κB activation in prostate cancer cells and thus overcomes TRAIL resistance [115]. FIS and cabazitaxel increase PARP cleavage, also increase pro-apoptotic protein Bax levels, and lower anti-apoptotic Mcl-1 levels. Modulation of Bax and Mcl-1 expression resulted in an increased Bax: Mcl-1 ratio in an apoptotic manner and the level of PCNA expression was inhibited in combined therapy [114].

As cancer is one of the most common causes of human morbidity and mortality worldwide, scientists are trying to discover new compounds and therapeutic approaches to reduce cancer cell survival. Cancer cells acquire a number of distinctive features, including the ability to proliferate in an exogenous manner independent of the growth-promoting signal, to invade surrounding tissues and form distant metastases. In addition, they promote angiogenesis, avoid mechanisms that limit cell proliferation (such as apoptosis and aging), and evade immune surveillance. All these activities reflect changes in signaling pathways that control apoptosis, cell cycle, proliferation, motility and cell survival (Figure 3) [127]. Numerous in vivo and in vitro studies have shown that flavonoids may be used as potential anticancer drugs because they affect cell signaling pathways [128]. Pharmacological targeting of the unregulated signaling pathways of oncogenic cells is a major challenge for scientists. Despite significant progress in understanding the mechanisms underlying cancer and parallel breakthroughs in identifying pharmacologically active natural products, much is still unknown [129]. In preclinical studies, FIS, found in many fruits and vegetables, has been shown to inhibit cancer growth without causing toxicity to normal cells. While the in vitro and in vivo data appear convincing, well-designed human clinical trials are needed to ultimately determine efficacy in a variety of tumors [130]. Of all the many biological effects of FIS, its anticancer potential has recently been studied, making it a promising agent in the prevention and treatment of cancer. We discussed here and in the previous work [8] that FIS is a functionally pleiotropic molecule, possessing many intracellular targets, influencing various cell signaling processes, usually altered in tumor cells. Influencing multiple pathways simultaneously may help kill cancer cells and slow the build-up of drug-resistance. Moreover, FIS shows synergistic anticancer activity in combination with some common clinical anticancer drugs, e.g., Gemcitabine [131], 5-fluoroucyl [47], paclitaxel [98] and TRAIL [115]. The combination of FIS and chemotherapeutic agents increases the chemosensitivity of cancer cells and reverses drug resistance.

## 3. Conclusions

Naturally occurring, common diet ingredients, in an original form or after chemical modification, have a vast potential to be implemented for clinical, therapeutic protocols in various fields of oncology. Undoubtedly, the relatively high cost of natural compound extraction and clinical trials involving such substances may hamper and limit their wider clinic indications. Pharmaceutical companies worldwide should invest in cutting-edge technologies supporting the use of phytochemicals in medicine, waiving the expected patents for low-income countries with high rates of neoplasm diseases. High quality, robust research plays a pivotal role in promoting their clinical application. Despite the relatively broad evidence of the impact of FIS on various tumor cells, detailed knowledge on the regulation of some signaling pathways does not exist, such as the activity of the TGF/SMAD signaling pathway. Transforming growth-factor beta (TGF) signaling is involved in a wide variety of cellular processes, such as differentiation, cell-cycle arrest, and immune regulation. TGF signaling has been linked to cancer, among others. TGF proteins bind to cell surface receptors, initiating a signaling cascade. We also have incomplete information on the potential of FIS in relation to JAK/STAT [90], sonic hedgehog [132] and NOTCH [133] signaling in various tumors.

Nevertheless, if FIS is shown to be associated with improved chemotherapy or radiotherapy outcomes, then clinicians will be able to exploit the synergistic effects of the combined protocols. This will result in the possibility of lowering the doses, and thus a reduction in toxicity. Inevitably new clinical trials are crucial to establish the complete chemopreventive and chemotherapeutic efficacy of FIS in adjuvant anticancer therapy. In the future, ultraprecise methods of plant gene editing may revolutionize the development of target-specific and “customized” molecules with a vast potential as anticancer agents. The utilization of artificial intelligence technologies to analyze big data associated with biological activities of phytochemicals might be a game changer in the biomedical science and pharmacological fields.

## Figures and Tables

**Figure 1 nutrients-14-02604-f001:**
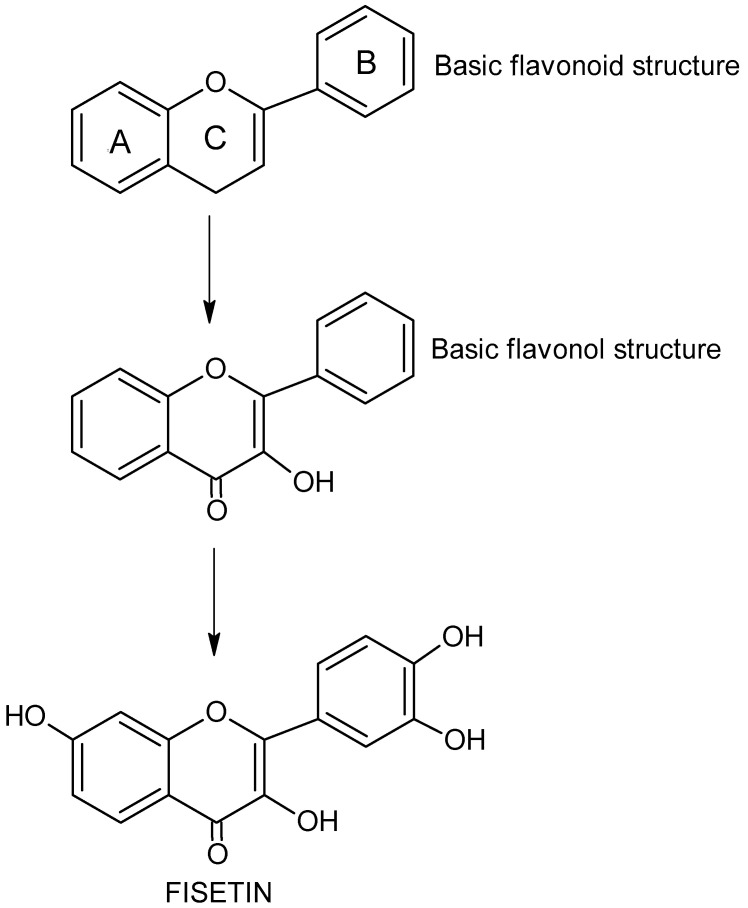
The chemical structure and formula of flavonoids, flavonols and fisetin.

**Figure 2 nutrients-14-02604-f002:**
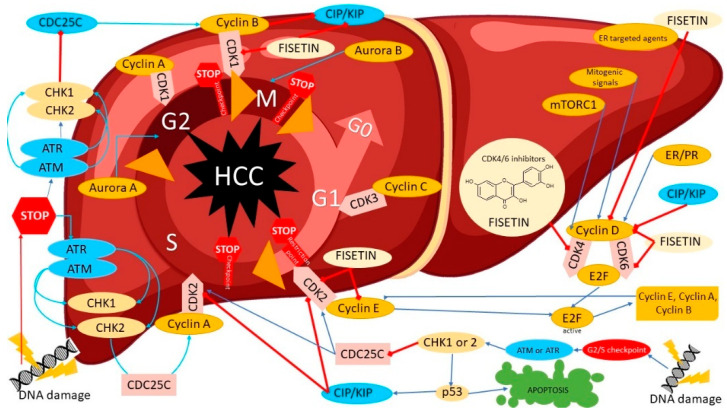
The potential mechanisms of cell-cycle inhibition of liver cancer cells (hepatocellular cancer, HCC) by modulation of several regulatory proteins under the influence of fisetin such as CDK4, Cyclin-dependent kinase 4; CDK6, Cyclin-dependent kinase 6; Cyclin D; CDK2, Cyclin-dependent kinase 2; CIP/KIP, the Cip/Kip family, namely, p21(Cip1), p27(Kip1) and p57(Kip2); CDK1, Cyclin-dependent kinase 1; CDC25C, M-phase inducer phosphatase 3; CDK3, Cyclin-dependent kinase 3; CHK1, Checkpoint kinase 1; CHK2, Checkpoint kinase 2; ATM, serine/threonine kinase; ATR, Serine/threonine-protein kinase; AuroraB, Aurora kinase B; AuroraA, Aurora kinase A; E2F, E2 transcription factors.

**Figure 3 nutrients-14-02604-f003:**
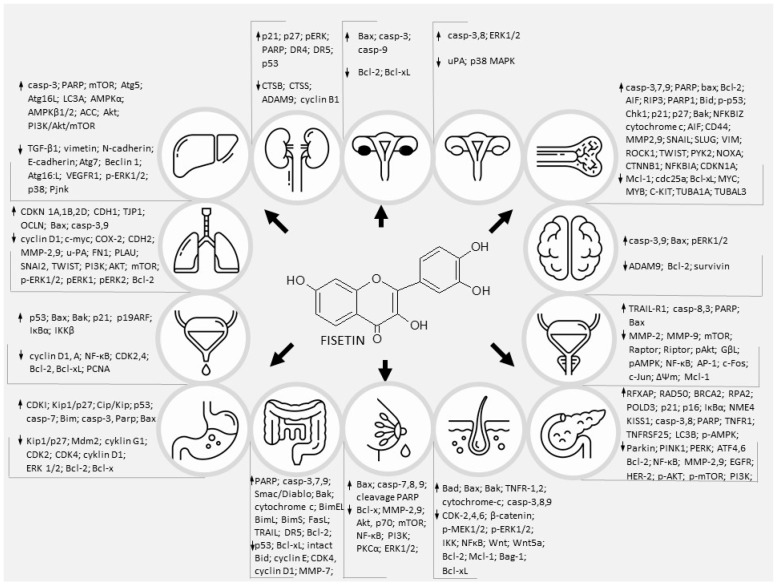
The multilevel, ultracellular effects of fisetin on major signalings that are involved in various cancers.

**Table 1 nutrients-14-02604-t001:** Anticancer mechanisms of fisetin.

Organs	Cell Line/Animals	IC50 [µM]	Mechanism	Ref.
Bladder	Wistar rats	Nd	FIS increases in the number of TUNEL-positive cells FIS regulates the expression of apoptosis-related proteins FIS downregulates NF-κB pathways FIS upregulates the expression of ARF	[28]
	T24	≈70 ^#^ ≈90 *	FIS inhibits the proliferation of T24 and EJ cells by inducing apoptosis and blocking cell-cycle progression in the G0/G1 phase FIS increases the expression of p53 and p21 proteins FIS decreases the levels of cyclin D1, cyclin A, CDK4 and CDK2 FIS increases the expression of Bax and Bak FIS decreased the levels of Bcl-2 and Bcl-xL FIS triggers mitochondrial apoptotic pathway	[27]
	EJ	>100 * ≈80 ^#^		
	J82	>100 * ≈80 ^#^		
Breast	SUM159 MDA-MB-468	ND	FIS influences MAPK/ERK pathway to impair RNA Pol I activity and rRNA biogenesis FIS localizes to the nucleolus and reduces the number of nucleoli per cell FIS affects RNA polymerase I activity and rRNA biogenesis FIS downregulates MAPK signaling FIS inhibits functional attributes of malignant mammary tumor cells	[18]
	HCC1806, HCC70, HCC1937, BT-549, 20 Hs578T, MDA-MB-231, 157, 468	ND	FIS inhibits migration of metastatic TNBC cells FIS inhibits migration and matrix invasion of TNBC cells FIS inhibits of metastasis in zebrafish FIS targets different components and substrates of the oncogenic PI3K/AKT pathway and reduces their activities FIS disrupts activities of several protein kinases in MAPK and STAT pathways	[87]
	MDA-MB-453	ND	Fisetin induces apoptosis in HER2/neu-overexpressing breast cancer cells Fisetin increases PI3K activity at 10 µM, which gradually declines on treatment with higher concentrations (>25 µM) FIS (10 µM) increases phosphorylation of Akt in MDA-MB-453 cells FIS decreases tyrosine phosphorylation of HER2 FIS decreases the levels of HER2/neu	[31]
	4T1 JC	ND	FIS inhibits cell migration and colony formation FIS decreases MMPs production and increases HO-1 expression Nrf2 mediates FIS-induced HO-1 expression in breast cancer cells	[33]
	4T1	≈80 * ≈40 ^#^	FIS inhibits breast cancer cell viability FIS inhibits the proliferation, migration and invasiveness of mammary carcinoma cells FIS induces the apoptosis of mammary carcinoma cells FIS regulates the PI3K/Akt/mTOR pathway in 4T1 mammary carcinoma cells FIS inhibits the primary tumor growth of 4T1 cells	[32]
	MCF-7	≈35 * ≈35 ^#^		
	MDA-MB-231	>100 * >100 ^#^		
	MDA-MB-231	≈100 *	FIS suppresses the proliferation, migration and invasion FIS reverses EMT in TNBC Cells FIS suppresses PI3K-Akt-GSK-3β signal pathway but upregulated PTEN expression	[88]
	BT549	≈100 *		
	MDA-MB-231	ND	FIS causes inhibition of cell growth in MDA-MB-231 breast cancer cells	[89]
	MDA-MB-468	>100 * 100 ^#^	FIS inhibits breast cancer cell growth FIS inhibits TNBC cell division and cell-cycle progression FIS causes tnbc cells to undergo apoptosis FIS inhibits histone h3 phosphorylation FIS disrupts the mitochondrial membrane and causes caspase activation in tnbc cells	[34]
	MDA-MB-231	>100 * >100 ^#^		
	MCF-7	ND	FIS attenuates TPA-induced cell invasion in MCF-7 cells FIS inhibits the activation of the PKCα/ROS/ERK1/2 and p38 MAPK signaling pathways	[35]
	MCF-7	≈40 *	FIS exhibits substantial cytotoxicity in caspase-3-deficient MCF-7 cells FIS does not induce necroptosis in MCF-7 cells FIS induces caspase-dependent cell death in MCF-7 cells FIS induces mitochondrial depolarization and p53-independent cell death in MCF-7 cells FIS inhibits autophagy in MCF-7 cells	[36]
	MDA-MB-231	>100 *		
Brain	GBM8401	>100 *	FIS exhibits effective inhibition of cell migration and inhibits the invasion of GBM8401 cells FIS inhibits the expression of ADAM9 protein and mRNA FIS phosphorylates ERK1/2 in a sustained way	[39]
	T98G	93 * 75 ^#^	FIS upregulates the expression of caspase-3, caspase-9, caspase-8, and bax FIS downregulates the expression of Bcl-2 and survivin	[38]
Cervix	HeLa	ND	FIS induces morphological changes and inhibits proliferation FIS changes nuclear morphology FIS leads DNA fragmentation FIS encourages G2/M arrest and modulates cell-cycle regulatory genes FIS activates extrinsic and intrinsic pathways FIS modulates expression of various pro- and anti-apoptotic proteins FIS elevates caspase-3, caspase-8 and caspase-9 activity FIS changes the aberrant MAPK and PI3K/AKT/mTOR in HeLa cells	[90]
	HeLa	52 * 36 ^#^	FIS induces apoptosis of HeLa cells in a dose- and time-dependent manner FIS triggers the activations of caspases-3 and -8 and the cleavages of poly (ADP-ribose) polymerase FIS induces a sustained activation of the phosphorylation of ERK1/2 FIS significantly reduces tumor growth in mice with HeLa tumor xenografts	[44]
Colorectal	SW480, HCT116, HT29	>100 * >100 * >100 *	FIS reduces the expression of PI3K, phosphorylation of AKT, mTOR, its target proteins, constituents of mTOR signaling FIS increases the phosphorylation of AMPKα.	[47]
	LoVo OR-LoVo CPT11-LoVo	>100 * >100 * ≈50 *	FIS induces apoptosis in LoVo cells, OR-LoVo, and CPT11-LoVo cells FIS induces apoptosis and inhibits survival pathway in parental and chemoresistance colon cancer cells FIS inhibits tumor growth in nude mice	[52]
	Caco-2	≈30 *	FIS inhibits cellular proliferation and viability of colorectal cancer cell line FIS induces apoptosis FIS inhibits PGE2 production	[91]
	Mouse xenograft models	ND	FIS inhibits tumor growth in a mouse CT-26 xenograft model FIS induces p53 and suppresses securin protein expression	[92]
	HCT-116 HT-29	ND	FIS reduces the surviving cell fraction in p53 wild-type HCT116 cells FIS prolongs radiation-induced G2/M arrest and enhanced radiation-induced cell growth arrest in HT-29 cells FIS suppresses radiation-induced phosphorylation of H2AX and phospho-Chk2 (Thr-68) in HT-29 cells FIS enhances radiation-induced caspase-dependent apoptosis in HT-29 cells FIS enhances radiosensitivity of irradiated HT-29 cells via the inhibition of AKT-ERK pathways	[93]
	HCT116	≈540 * ≈140 ^#^ ≈130 ^^^	FIS induces growth inhibition of HCT116 and HT29 colon cancer cells FIS induces apoptosis of HT29 colon cancer cells FIS inhibits expression of COX2 in HT29 cells FIS inhibits COX2 promoter activity and PGE2 secretion FIS inhibits β-catenin pathway in HT29 cells FIS inhibits expression and translocation of TCF1 and TCF4 in HT29 cells FIS inhibits COX2 expression through downregulation of TCF4 FIS inhibits activation of EGFR in HT29 cells FIS inhibits activation of NF-κB in HT29 cells FIS reduces expression of Wnt target genes and inhibits colony formation	[50]
	HT-29	≈240 * ≈140 ^#^ ≈57 ^^^		
	COLO205, HCT-116, HCT-15, HT-29	ND	FIS with NAC increases the expression of cleaved caspase-3 and PAPR protein FIS with NAC produces greater density of DNA ladders NAC and FIS inhibits on ERK protein phosphorylation	[51]
	COLO205	>100 *	FIS inhibits cellular proliferation and viability on human COLO205 colon cancer cells in the presence and absence of the HSP90 inhibitors HSP90 inhibitors enhance FIS-induced cytotoxicity HSP90 inhibitors increase expression of cleaved caspase-3 and the PAPR protein Increased caspase-3 and caspase-9 activities were detected in cancer cells treated with FIS and HSP90 inhibitors	[53]
	HCT-116 HT-29	ND	FIS exhibits higher cytotoxicity in securin-null HCT116 cells Knockdown of securin expression in cells enhances FIS-induced cell death p53-deficient human colon cells are resistant to FIS-induced apoptosis and cytotoxicity	[48]
	HCT-116 HT-29	ND	FIS induces apoptosis of HCT-116 cells FIS induces depolarization of the mitochondrial membrane in HCT-116 cells FIS alters the levels of Bcl-2 family proteins FIS induces Bax translocation to mitochondria FIS increases cleavage of caspase-8	[49]
	HT-29	ND	FIS inhibits both cell growth and DNA synthesis FIS decreases the activities of cyclin-dependent kinases CDK2 and CDK4 FIS inhibits CDK4 activity	[54]
Kidney	786-O	≈50 *	FIS decreases RCC cell viability FIS induces cell-cycle arrest in the G2/M phase FIS inhibits migration and invasion FIS inhibited CTSB, CTSS, and ADAM9 FIS upregulates ERK activation	[57]
	CaKi-1	≈30 *		
	ACHN	≈40 *		
	A-498	≈40 *		
	Stem cells	ND	FIS inhibits HuRCSC cell division and proliferation, invasion, in vivo tumorigenesis and angiogenesis FIS decreases TET1 expression levels in HuRCSCs	[94]
	CaKi	ND	FIS induces apoptosis in Caki cells FIS induces sub-G1 population and cleavage of PARP FIS induced apoptosis through upregulation of DR5 expression FIS induces p53 protein expression FIS induces upregulation of CHOP expression and ROS production	[58]
Leukemia	K562	ND	FIS (from 10 to 50 µM) is not highly toxic to the K562 cells FIS did not cause any apparent changes in the viability FIS is not a potent inducer of apoptosis FIS-treated cells exhibits a greater capacity to invade than the untreated ones FIS treatment enhances the nuclear localization of β-catenin	[62]
	WEHI-3	ND	FIS decreases total viable cells through G0/G1 phase arrest and induced sub-G1 phase FIS induces cell apoptosis by the formation of DNA fragmentation FIS induces intracellular Ca^2+^ increase FIS decreases the ROS production and the levels of ΔΨm FIS increases the activities of caspase-3, -8, -9 FIS reduces expressions of cdc25a FIS increases expressions p-p53, Chk1, p21 and p27 FIS inhibits Bcl-2 and Bcl-xL and increases Bax and Bak	[64]
	HG-3, EHEB,	ND	FIS augments the cytotoxic activity of luteolin	[95]
	K562	≈160 ^#^ ≈120 ^^^	FIS inhibits growth of K562 cells FIS induces apoptosis of K562 cells FIS increases caspase-3 activity FIS arrests cell cycle at both S and G2/M phases	[63]
	HL60	≈80 ^#^ ≈45 ^^^	FIS triggers apoptosis in HL60 cells FIS induces loss of mitochondrial membrane potential FIS increases caspase-3 activity FIS arrests cell cycle at the G2/M phase	[65]
	THP-1	ND	FIS affects survival of acute monocytic leukemia cells FIS-treatment results in increase in NO levels FIS-treatment induces double strand DNA breaks FIS induces NO production downregulates mTOR activity and causes activation of caspases FIS alters Ca^2+^ levels and activates caspases	[67]
	K562	ND		
	U937	ND		
	HL-60	ND	FIS induces apoptosis of K562 cells FIS causes rapid and transient induction of caspase-3/CPP32 activity FIS does not cause caspase-1 activity FIS decreases procaspase-3 protein	[66]
Liver	HepG2	≈80 *	FIS performs as DR2 agonist to suppress liver cancer cells proliferation, migration and invasion FIS activates caspase-3 signaling to induce apoptosis FIS downregulates VEGFR1, p-ERK1/2, p38 and pJNK	[70]
	HCC-LM3	≈40 *		
	SMMC-7721	>100 *		
	Charles foster rats	ND	FIS normalizes the enhanced expression of TNFα and IL1α	[96]
	HepG2	ND	FIS inhibits autophagy by the activation of PI3K/Akt/mTOR and modulation of AMPK signaling pathways. FIS inhibits autophagic flux in HepG2 cells. FIS inhibits autophagy through AMPK regulation FIS exposure does not show any significant ATP level changes	[71]
	HepG2	ND	FIS decreases cell viability and proliferation of HepG2 cells FIS induces cell-cycle arrest in the G2/M phase FIS induces both apoptosis and necroptosis in HepG2 cells FIS induces ROS production FIS causes a marked increase in the expression of TNFα and IKκB FIS causes a marked decrease in NF-κB, pNF-κB and pIKκB expression FIS reduces the expression of Bcl2, and elevates levels of Bax, caspase-3, and PARP	[72]
	SK-HEP-1	ND	FIS shows dose-dependent cytotoxic effects on SK-HEP-1 cells, accompanied by DNA fragmentation FIS induces cellular swelling and the appearance of apoptotic bodies FIS induces of apoptosis in SK-HEP-1 cells FIS activates Caspase-3 signaling to induce apoptosis	[73]
Lung	H1299 A549	ND	FIS decreases the expression of signaling proteins (β-catenin, NF-κB, EGFR, STAT-3) FIS decreases the ability of H1299 cells to form colonies and potentiates the cytotoxic effects of tyrosine kinase inhibitor-erlotinib	[97]
	A549	ND	FIS inhibits A549 cell proliferation FIS causes cell-cycle arrest in A549 cells FIS induces apoptosis of A549 cells FIS suppresses cell adhesion, invasion and migration FIS inhibits the activation of the ERK signaling pathway via MEK1/2	[76]
	NCI-H460	ND	FIS increases the ER stress signaling FIS increases the level of mitochondrial ROS FIS induces mitochondrial Ca^2+^ overloading and ER stress FIS induced ER stress-mediated cell death via activation of the MAPK pathway	[78]
	A549	ND	FIS acts synergistically with paclitaxel to decrease the viability of A549 cells FIS synergized with PTX or ATO in A549 cells as well as that the synergistic effect of FIS and PTX was cell line-specific FIS induces autophagy in A549 cells	[98]
	A549	ND	FIS inhibits the adhesion, invasion, and migration in A549 cells FIS inhibits the expressions of MMP-2 and u-PA FIS inhibits the phosphorylation of ERK FIS inhibits the protein expressions of MMP-2 and u-PA FIS inhibits the DNA binding activities of NF-κB, c-Fos, and c-Jun	[79]
	A549	ND	FIS enhances chemotherapeutic effect of Cisplatin FIS reverses Cisplatin-resistance of cells through MAPK/Survivin/Caspase pathways	[99]
	Swiss albino mice	ND	FIS is a very successful drug in combating the mitochondrial dysfunction in an experimental model of lung carcinogenesis	[100]
	Swiss albino mice	ND	FIS significantly reduces the degree of histological lesions FIS restores the levels of lipid peroxidation (LPO), enzymic and nonenzymic antioxidants	[101]
Melanoma	WM35	ND	FIS inhibits YB-1 in mutant BRAF melanoma cells FIS binds to RSK and suppresses its kinase activity FIS induces modulation of YB-1/RSK signaling FIS suppresses YB-1/RSK signaling independent of its effect on ERK FIS reduces MDR1 levels	[102]
	A375	ND		
	M17	≈60 *	FIS decreases cells viability FIS induces apoptosis through the intrinsic pathway FIS damages MTP in uveal melanoma cells FIS increases the release of cytochrome c in cytosol FIS increases caspase-9 and -3 activities	[83]
	SP6.5	≈85 *		
	A375	ND	FIS reduces human melanoma cell invasion by inhibiting EMT FIS inhibits cell proliferation and tumor growth by downregulating the PI3K pathway	[103]
	A375, RPMI-7951	ND	FIS treatment inhibits PI3K signaling pathway in melanoma cells FIS enhances sorafenib-mediated cleavage of caspase-3 and PARP FIS modulates expression of Bcl2 family proteins in BRAF-mutated melanoma cells FIS with sorafenib effectively down-regulates MAPK and PI3K signaling pathways FIS potentiates the sorafenib-mediated tumor growth inhibition in athymic nude mice	[104]
	A375, RPMI-7951, Hs294T	ND	FIS reduces invasion of melanoma cells FIS inhibits invasion of melanoma cells in three-dimensional human skin equivalents FIS inhibits melanoma cell invasion by targeting MEK1/2 and NFκB FIS inhibits phoshorylation of MEK1/2 and ERK1/2 FIS inhibits nuclear translocation of NFκB	[86]
	Mel 928	ND	FIS decreases the viability of 451Lu cells FIS induces G1-phase arrest in 451Lu cells FIS downregulates of Wnt protein and its coreceptors FIS decreases nuclear β-catenin levels FIS interferes with the functional cooperation between TCF-2 and β-catenin	[85]
	WM35	ND		
	451Lu	80 * ≈37 ^#^ ≈18 ^^^		
Stomach	AGS	≈45 * ≈13 ^#^	FIS decreases the viability FIS induces apoptosis	[105]
	SGC7901	ND	FIS inhibits proliferation of gastric cancer cell and induces apoptosis FIS increases the proportion of cells at G2/M phase with simultaneous reduction in cells at S phase FIS increases caspase-7 activities FIS reduces the expression of Bcl2, Bcl-x and Bid FIS reduces of the activation of ERK 1/2	[106]
	AGS	ND	FIS inhibits cell proliferation, growth and viability FIS induces a G1 phase arrest in gastric cancer cells FIS increases the level of cyclin-dependent kinase inhibitor (CDKI) Cip1/p21 FIS induces apoptosis and mitochondrial membrane depolarization FIS causes upregulation of total p53 and its activation by phosphorylation at S15 position FIS increases the phosphorylation of gamma-H2A.X S139 in both the cell lines	[107]
	SNU-1	ND		
Ovary	A2780	ND	FIS reduces cell growth in both OC cell lines FIS induces apoptosis and necroptosis FIS-induced cell necroptosis involves the RIP3/MLKL pathway.	[108]
	VOCAR-3	ND		
	A2780	ND	FIS and CIS effectively inhibit proliferation of A2780 cells FIS induces nuclear fragmentation of A2780 cells	[109]
Pancreas	PANC-1	ND	FIS promotes DSBs and inhibits HR repair in pancreatic cancer cells FIS regulates PHF10 expression via m6A RNA modification	[110]
	PANC-1	ND	FIS dampens the proliferation of pancreatic cancer by downregulating Ki67 expression FIS triggers apoptosis of human pancreatic cancer cells FIS reduces the ability of infiltration and migration FIS dampens the expression of EMT-linked proteins FIS dampens the PI3K/AKT/mTOR axis FIS dampens pancreatic tumor growth of cell xenografts in nude mice	[5]
	BxPC-3	≈120 * ≈75 ^#^	FIS inhibits the viability of human pancreatic cancer cells FIS induces S phase and DNA damage in pancreatic cancer cells FIS inhibits cell proliferation and induces DNA damage FIS upregulates of expression of RFXAP and other DNA-damage response genes FIS induces DNA damage via RFXAP/KDM4A-dependent histone H3K36 demethylation	[111]
	MiaPACA-2	ND		
	PANC-1	≈400 * ≈200 ^#^		
	HPC-Y5	ND		
	PANC-1	≈350 * ≈300 ^#^	FIS inhibits the viability of human pancreatic cancer cells FIS induces apoptosis and autophagy FIS stimulates the AMPK pathway FIS induces ER stress	[112]
	BxPC-3	ND		
	AsPC-1	>100 * ≈40 ^#^	FIS induces apoptosis FIS inhibits invasion of chemoresistant PaC AsPC-1 cells through suppression of DR3 mediated NF-κB activation	[113]
Prostate	PrEC	ND	FIS and cabazitaxel significantly suppresses colony formation FIS increases cleavage of PARP FIS increases the level of Bax FIS decreases the level of Mcl-1	[114]
	22Rν1	ND		
	C4-2	ND		
	LNCaP, PC3 DU145	ND	FIS sensitizes the TRAIL-resistant androgen-dependent LNCaP cancer cells FIS augments TRAIL-mediated cytotoxicity and apoptosis in prostate cancer LNCaP cells FIS increases the expression of TRAIL-R1 FIS decreases the activity of NF-κB	[115]
	PC3, DU145 LNCaP	ND	FIS induces growth inhibition of PC3 cells FIS decreases the activity of mTOR kinase FIS inhibits phosphorylation of mTOR FIS inhibits expression of the mTORC1 and mTORC2 constituents FIS inhibits formation of mTORC1/2 FIS inhibits activation of Akt FIS activates mTOR inhibitor tuberous sclerosis complex 2	[116]
	PC-3	ND	FIS inhibits the adhesion, invasion, and migration in PC-3 cells FIS inhibits the expressions of MMP-2 and MMP-9 FIS inhibits the phosphorylation of JNK and Akt in PC-3 cells FIS inhibits the DNA binding activities of NF-κB, c-Fos and c-Jun	[117]

ND—not determined due the results obtained; *—IC50 values at 24 h; ^#^—IC50 values at 48 h; ^^^—IC50 values at 72 h.

## Data Availability

Not applicable.

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
