# Peer review of "Fisetin, a Potent Anticancer Flavonol Exhibiting Cytotoxic Activity against Neoplastic Malignant Cells and Cancerous Conditions: A Scoping, Comprehensive Review"

_nutrients, 2022, doi:10.3390/nu14132604_

Round 1
Reviewer 1 Report
In this manuscript, the authors examine the anticancer effect of flavonol and analyze its potential to be used as a cancer treatment, used alone or in combination. Overall, I found the review well prepared, scientifically sound, complete and well organized.
Author Response
Answers to Reviewer 1:
We would like to express our gratitude for the detailed and critical reading of the manuscript. We are very grateful to the Reviewer for the very positive comments. The revised manuscript has been addressed accordingly to the as the other reviewer's suggestions.
We strongly believe that our updated manuscript will draw the great interest of the readers of Nutrients journal.

Reviewer 2 Report
-Before conclusion, make a separate section and/or draw a summary schematic diagram of the major anti-cancer effects of fisetin on major signalings that are involved in various cancers.
Author Response
Answers to Reviewer 2:
First of all, I would like to thank you for taking time out of your busy schedule to review this manuscript. Your positive comments and suggestions have important guiding significance to my scientific research work.
Comment 1: -Before conclusion, make a separate section and/or draw a summary schematic diagram of the major anti-cancer effects of fisetin on major signalings that are involved in various cancers.
Thank you for your comments. The conclusions chapter has been subtly revised. Additionally, the place of Figure 2 has been changed and it has been placed before Conclusion.
All changes suggested by the Reviewer were introduced to the corrected version of the text. Finally, thank you again for your valuable advices.
